# Explainable Image Quality Analysis of Chest X-Rays

**Caner Özer** [1]                       OZERC@ITU.EDU.TR

**İlkay Öksüz** [1,2]                    OKSUZILKAY@ITU.EDU.TR

[1] *Department of Computer Engineering, Istanbul Technical University, Turkey*

[2] *School of Biomedical Engineering & Imaging Sciences, King's College London, U.K.*

## Abstract

Medical image quality assessment is an important aspect of image acquisition where poor-quality images may lead to misdiagnosis. In addition, manual labelling of image quality after the acquisition is often tedious and can lead to some misleading results. Despite much research on the automated analysis of image quality for tackling this problem, relatively little work has been done for the explanation of the methodologies. In this work, we propose an explainable image quality assessment system and validate our idea on foreign objects in a Chest X-Ray (Object-CXR) dataset. Our explainable pipeline relies on NormGrad, an algorithm, which can efficiently localize the image quality issues with saliency maps of the classifier. We compare our method with a range of saliency detection methods and illustrate the superior performance of NormGrad by obtaining a Pointing Game accuracy of 0.862 on the test dataset of the Object-CXR dataset. We also verify our findings through a qualitative analysis by visualizing attention maps for foreign objects on X-Ray images.

**Keywords:** Saliency detection, Image Quality Analysis, X-Ray, Foreign Object Detection, NormGrad

## 1. Introduction

High medical image quality is essential for extracting clinically meaningful information from medical images. Image artefacts cause degradation of medical image quality CT, (Ma et al., 2020), MRI (Barrett and Keat, 2004) and for X-Ray images (Veldkamp et al., 2009) due to various factors. Apart from problems, whose effects can be directly evaluated through objective metrics such as Peak Signal-to-Noise Ratio or Structural Similarity Index, there are additional image quality issues (e.g. foreign objects inside, breathing artefacts), that require additional manual annotation for automatic detection. There is a need of manual labelling of these objects, which is a lengthy procedure and subject to human errors.

Chest X-Ray is one of the most widely used imaging platforms for disease diagnosis including tuberculosis (Liu et al., 2017) and COVID-19 (Oh et al., 2020). Unfortunately, patients undergoing Chest X-Ray may carry some foreign objects such as buttons or clips on themselves. These result in the foreign object appearance on the images (JFHealthcare, 2020), which makes it difficult to diagnose. In this regard, an automated system that can detect the existence of foreign objects would accelerate the whole image acquisition procedure and enable the reacquistion of these images on the spot. A naïve way to overcome the aforementioned issues is to train a classifier which aims to distinguish between the good and poor quality images. However, the classifier may generate some errors in the form of misclassified images, and interpretation of such errors holds the key in improving the detection accuracy. Unexplainability of the performance of automatic quality assessment techniques challenge the potential clinical translation of such methods.

Current state-of-the art solutions in computer vision literature resort to visual explanations either by in-model or post-hoc explanation methods (Singh et al., 2020). On one hand, in-model methods such as attention maps (Schlemper et al., 2019) integrate the interpretability method within the model itself. On the other hand, post-hoc explanation methods use a pre-trained model to produce saliency maps in order to demonstrate what the model learned as a result of its training procedure. In this regard, Grad-CAM (Selvaraju et al., 2020) uses the gradient-weighted class activation maps to construct a final saliency map. Meanwhile, Score-CAM (Wang et al., 2020) comes up with an alternative scheme for activation map weighting in contrast to Grad-CAM. Guided Backpropagation (Guided BP) (Springenberg et al., 2015) and Input x Gradient (Input x Grad) (Shrikumar et al., 2017) directly use the gradient to construct a saliency map. In addition, there also exist a number of methods (Oh et al., 2020; Schut et al., 2020), which leverage counterfactual analysis, to explain a classifier's decision. As a result, it is possible not only to make a prediction, but it can also provide an explanation to the clinician about the regions that deteriorates the image quality or by pointing the relevant regions which show the disease patterns. For example, (Joshi et al., 2020) have proposed a novel segmentation-guided model explanation framework for recognizing Diabetic Macular Edema from Optical Coherence Tomography imaging. Also, (Costa et al., 2017) proposed an end-to-end explainable image quality analysis framework on retinal images.

In this paper, we plan to utilize NormGrad (Rebuffi et al., 2020) for the detection of saliency maps and illustrate the superior performance of this technique when highlighting the underlying decision for foreign object detection of Chest X-Rays. NormGrad is based on aggregating multiple saliency maps by using Frobenius Norm and introduces a precise saliency map instead of a weighted summation of Grad-CAM. We show that NormGrad provides more accurate saliency maps comparing other well-known methods for saliency extraction on Chest X-Rays, where we verify this claim through qualitative and quantitative analysis. To the best of our knowledge, this work is the first that uses NormGrad in the medical image quality analysis, in addition to being the first paper by performing an explainable image quality analysis on Chest X-Ray data.

## 2. Method

In this section, we briefly describe Grad-CAM (Selvaraju et al., 2020) and NormGrad frameworks (Rebuffi et al., 2020). These methods generally aid us to construct the saliency maps of deep neural network based classifiers.

For both of these frameworks, we assume that there exists a pre-trained neural network in which we would like to extract the knowledge about some target layer $k_t$. We also define its preceding layers with $p$ and succeeding with $q$. Given an input image $\mathbf{x} \in \mathbb{R}^{C \times H \times W}$, where $C$ refers to the number of input channels and $H$ and $W$ correspond to the size of the image, we can also define $\mathbf{x}^{in} \in \mathbb{R}^{K \times H' \times W'}$, $\mathbf{x}^{out} \in \mathbb{R}^{K' \times H' \times W'}$, and the network output, $\mathbf{y}$, such that

$$\mathbf{x}^{in} = p(\mathbf{x})$$
$$\mathbf{x}^{out} = k_t(\mathbf{x}^{in}) \tag{1}$$
$$\mathbf{y} = q(\mathbf{x}^{out}).$$

In order to run Grad-CAM, the gradient w.r.t. the parameters of layer $k_t$, $\mathbf{g}^{out} \in \mathbb{R}^{K' \times H' \times W'}$, is needed to be accumulated alongside the activations of the same layer, $\mathbf{x}^{out}$. Unless stated otherwise, the gradient is calculated by assuming that $\mathbf{y}$ is the ground-truth class label while this may not hold to be true for all samples.

In addition, consider that a virtual identical layer, $\tilde{k}_t$, is present right after the layer $k_t$, whose output is $\tilde{\mathbf{x}}^{out}$ and satisfies the property $\tilde{\mathbf{x}}^{out} = \mathbf{x}^{out}$. The purpose of adding this layer is to assure that activations and gradients are collected from the same position of the network as in other saliency methods. Moreover, this layer can be any type among bias, scaling or convolutional layers. If the framework choice is NormGrad, we accumulate the output activations, $\tilde{\mathbf{x}}^{out}$, and the corresponding upstream gradient, $\mathbf{g}^{out}$, of the layer $\tilde{k}_t$.

## 2.1. Grad-CAM

In general, Grad-CAM constructs the saliency maps in four steps. First, Grad-CAM creates the importance weight vector, $\alpha$, by

$$\alpha = \frac{1}{H \times W} \sum_{h,w} g^{out} \tag{2}$$

where $H \times W$ acts as an averaging term across all pixels. In addition, this weight vector provides the importance of the filters in $k_t$. After two operations take place between $\alpha$ vector and $\mathbf{x}_{out}$ as in Equation (3)

$$S = \sum_k \alpha_k \odot \mathbf{x}_k^{out} \tag{3}$$

where each scalar component in $\alpha$ and each matrix component in $\mathbf{x}^{out}$ are multiplied and summated in order to obtain an aggregated spatial contribution, $S$. However, as a consequence, there may exist some inhibited regions with some value less than 0. In this regard, Grad-CAM saturates the inhibited regions by using ReLU activation function on the aggregated spatial contribution in which we obtain a final heat-map $\mathbf{m}$ with a shape of $[H' \times W']$ as in Equation (4).

$$\mathbf{m} = ReLU(S) \tag{4}$$

Finally, Grad-CAM up-samples $\mathbf{m}$ to the original input image size which is $[H \times W]$.

## 2.2. NormGrad

In contrast to Grad-CAM, NormGrad exploits $\mathbf{x}^{\tilde{out}}$ and $\mathbf{g}^{out}$ directly without estimating a weight vector from the gradients or activations. Moreover, it uses in-place virtual identity layers to represent different ways of obtaining spatial contributions. In Table 1, we provide

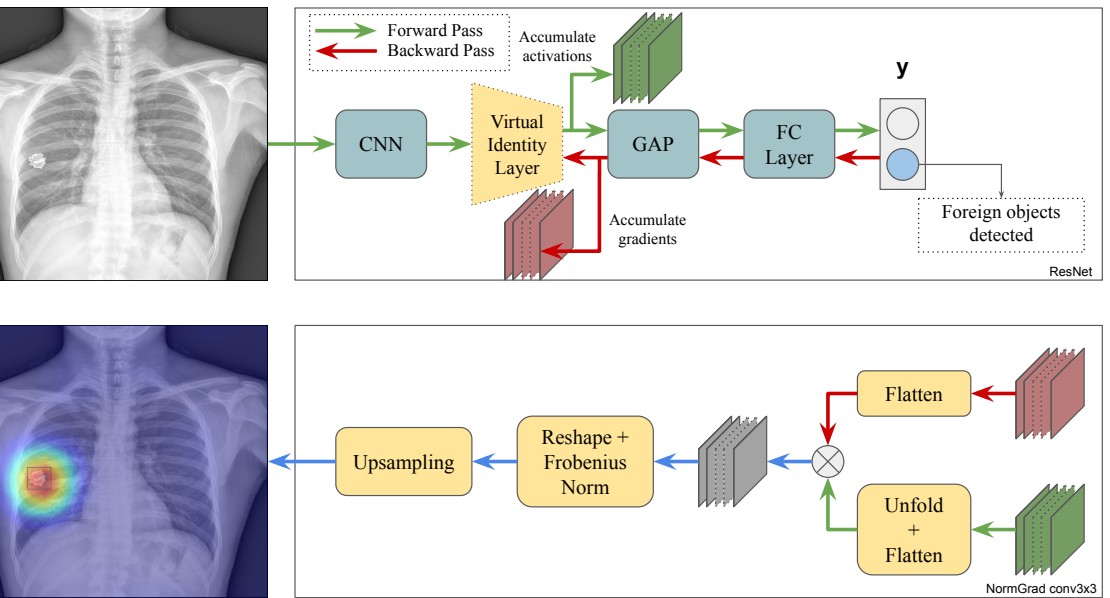

Figure 1: NormGrad Framework. The top figure shows the flow of a neural network where a virtual identity layer is placed right after the last convolutional layer to accumulate the activations and gradients. The bottom figure demonstrates how they are used to obtain a unified heat map by using NormGrad. Red and blue colors within the heat maps point to regions with high and low values. CNN, GAP and FC stand for Convolutional Neural Networks, Global Average Pooling and Fully-Connected, respectively.

the analytic derivations of the spatial contributions and their corresponding matrix/tensor sizes, depending on the virtual identity layer of choice. The first row corresponds to a virtual bias layer, which is directly equal to the upstream gradient, $\mathbf{g}^{out}$, and the second row corresponds to a virtual scaling layer that is equal to the element-wise product of activations and gradients. However, some additional operations are needed to be used for calculating the spatial contribution of a $N \times N$ convolutional layer. Suppose that we express the convolution operation using matrix multiplication as

$$\tilde{\mathbf{X}}^{out} = \tilde{\mathbf{W}}\mathbf{X}_{N \times N}^{out}, \tag{5}$$

where $\tilde{\mathbf{W}} \in \mathbb{R}^{K' \times N^2 K}$ denotes the parameters of the layer, $\tilde{k}_t$, $\mathbf{X}_{N \times N}^{out} \in \mathbb{R}^{N^2 K' \times H' W'}$ is the unfolded version of $\mathbf{x}_{out}$, and $\tilde{\mathbf{X}}^{out} \in \mathbb{R}^{K' \times H' W'}$ is the output of the convolution operation. The unfold operation is used to extract $N \times N$ patches from $\mathbf{x}_{out}$ which is mainly used to speed-up gradient calculation procedure. Each column of $\mathbf{X}_{N \times N}^{out}$ is denoted by $\mathbf{x}_{u,N \times N}^{out} \in \mathbb{R}^{N^2 K'}$ and used in the gradient of loss w.r.t. $\tilde{\mathbf{W}}$ that

$$\frac{dL}{d\tilde{\mathbf{W}}} = \sum_{u \in \Omega} \frac{d}{d\tilde{\mathbf{W}}} \langle \mathbf{g}_u^{out}, \tilde{\mathbf{W}}\mathbf{x}_{u,N \times N}^{out} \rangle = \sum_{u \in \Omega} \mathbf{g}_u^{out} \mathbf{x}_{u,N \times N}^{out \mathsf{T}}. \tag{6}$$

Table 1: Spatial contributions, shapes and saliency map formulas of different virtual identity layer choices.

| Layer | Spatial Contribution | Shape | Saliency Map |
|---|---|---|---|
| Bias | $\mathbf{g}_u^{out}$ | $K'$ | $\left\|\mathbf{g}^{out}\right\|$ |
| Scaling | $\mathbf{g}_u^{out} \odot \mathbf{x}_u^{out}$ | $K'$ | $\left\|\mathbf{g}^{out} \odot \mathbf{x}^{out}\right\|$ |
| Conv $N \times N$ | $\mathbf{g}_u^{out}\mathbf{x}_{u,N\times N}^{out\mathsf{T}}$ | $K' \times N^2 K'$ | $\left\|\mathbf{g}^{out}\right\| \left\|\mathbf{x}_{N\times N}^{out}\right\|$ |

The next step is to aggregate these spatial contributions in order to obtain a unified heat-map, $\mathbf{m}$, which has a shape of $[H' \times W']$. NormGrad uses $L^2$/Frobenius Norm as the aggregation function aiming to effectively handle the shape of each of the spatial contributions of a convolutional layer, which is a matrix. In the last column of Table 1, we show the analytical formulas for saliency maps to be generated by using $L^2$ Norm for bias and scaling layers, and Frobenius Norm for the Conv $N \times N$ layers. Following this aggregation procedure, NormGrad up-samples $\mathbf{m}$ to the original image size, $[H \times W]$ and obtain our final output for the single layer scenario of NormGrad.

In addition, it is also possible to add more than a single virtual identity layer inside the network, and combine all of the saliency maps generated by using the gradients and activations of these layers. In this paper, we selected uniform combination setting for heat-map combination which calculates the geometric mean of the given heat-maps. Additionally, we use the same type of virtual identity layers which ensures a fair assessment of performance. Given $J$ heat-maps prior to aggregation, we can obtain the combined heat-map, $\mathbf{M}$, such by,

$$\mathbf{M} = \Pi_{j=1}^{J} \sqrt[J]{\mathbf{m}_j}. \tag{7}$$

$\mathbf{M}$ will be the final output for the combined layer scenario of NormGrad.

## 3. Experimental Results

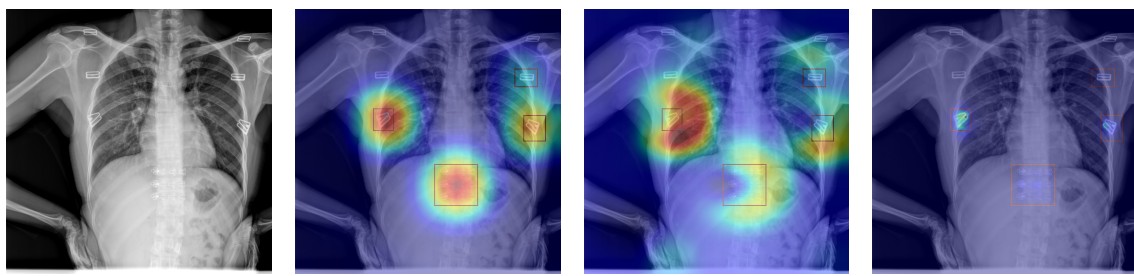

Figure 2: Attention maps of an image when there are multiple foreign objects to be detected. From left to right: Original Image, NormGrad conv1x1 single, Grad-CAM, NormGrad conv1x1 combined. Foreign objects outwards the lungs have not been annotated with a bounding-box in this dataset. (Best viewed zoomed in color)

We perform our experiments on Object-CXR dataset (JFHealthcare, 2020) which is a benchmarking dataset for foreign object recognition and localization on Chest X-Ray images. This dataset consists of 10,000 Chest X-Ray images in total, which 5,000 of them include foreign objects and 5,000 of them do not. We make the code and the experiments available on **https://github.com/canerozer/explainable-iqa**.

### 3.1. Image Quality Classification with ResNet-34 model

Our trained model for this benchmarking is a ResNet-34 model which takes a $600 \times 600 \times 3$ input image and recognizes whether there is at least a single foreign object in the image or not. We fine-tune the ResNet-34 model, which was pre-trained on ImageNet dataset (Deng et al., 2009), for 20 epochs using a batch size of 16 and cross-entropy loss function. In this regard, we duplicated the channel axis of the input image for 3 times, replaced the last layer of the ResNet-34 model, which now has 2 output neurons, and inherited the remaining layers from the pre-trained model. The optimization function is stochastic gradient descent with momentum where its learning rate is defined as 0.005. We also reduce the learning rate by dividing it to 10 in every 5 epochs. Lastly, we use color jittering, affine transformations and horizontal flips as the choice of data augmentations during training. With this model, we achieve an AUC score of 0.937 on the testing split of the Object-CXR dataset.

### 3.2. Experimental Setup for attention maps

We present our qualitative and quantitative analysis on the validation and testing splits of the Object-CXR dataset with 1,000 images for each split, which includes 500 images containing foreign objects and 500 of them do not. In this manner, we first identicate the differences in attention maps, when we modify the virtual identity layer type of the NormGrad framework. We do not only provide the results for a single heat-map but also we include the results when we combine 4 different heat-maps as well. Then, we compare the best performing NormGrad settings with other baselines such as Grad-CAM (Selvaraju et al., 2020), Guided Grad-CAM (Selvaraju et al., 2020), Guided Backpropagation (Springenberg et al., 2015) and Input x Gradient (Shrikumar et al., 2017). In order to provide a fair comparison among methods, we intentionally use the last convolutional block of ResNet-34, namely, layer4.2 for both Grad-CAM and single-layer setting of NormGrad. In addition, we also examine the combined-layer setting of NormGrad by involving layer2.0, layer3.0, layer4.0, and layer4.2 of ResNet-34. Qualitative results of NormGrad are obtained by using the virtual conv1x1 layer.

### 3.3. Qualitative Results

Figure 2 demonstrates an example with multiple foreign objects to compare the outputs of NormGrad and Grad-CAM. We notice that Grad-CAM provides attention maps on the target objects of interest with some offset. We define them as *skewed* attention maps, which are characterized by some circular sectors outwards the target objects. Furthermore, some salient regions do not intersect with any target objects, and for this reason, they are considered false positives. In stark contrast, a significant reduction in *skewness* of the attention maps on the target objects and fewer false positives suggest that the single-layer setting of NormGrad focuses more accurately than Grad-CAM on the target foreign

objects. Besides, when we use the combined-setting of NormGrad, we notice a sharp fall in the area of activity since earlier layers inhibit the heat-map. However, this condition may also completely suppress the heat-map activity on some target objects like the one at the top-right. Nevertheless, the combined heat-map is promising for a potential weakly-supervised semantic segmentation task of segmenting the foreign objects. It is because of NormGrad providing more reliable and precise attention maps than Grad-CAM.

In Figure 6 in Appendix D, we present a representative example from the Object-CXR dataset when there exists a misclassification by the model. Although our ResNet-34 model has not predicted any foreign objects in this image, the ground-truth label suggests the reverse. Consequently, it is possible to observe this misclassification error within the attention map of Grad-CAM. This method points to all regions except for the region represented by the bounding box. Hence, Grad-CAM is not robust enough to handle the misclassification errors. In stark contrast, NormGrad overcomes this problem by attending only to the foreign object.

### 3.4. Quantitative Results

We perform a quantitative analysis on the Pointing Game (Zhang et al., 2016) which aims to detect whether saliency maps align with the ground-truth bounding boxes. In Pointing Game, if the location of the maximum value of a saliency map is close to either one of the bounding box annotations with a pixel offset value, $\tau$, the saliency map is considered to be accurate. By defining these accurate saliency maps with $T$ and those who not with $F$, an accuracy metric $A$ is defined such that $A = \frac{T}{T+F}$. In Table 2, we report the quantitative results for methods of comparison and the ablation study of the proposed NormGrad setup for $\tau = 25$. We see that almost all of the NormGrad settings outperform other baseline settings in both validation and testing splits of Object-CXR with maximum accuracies of 0.880 and 0.862, respectively. The difference of NormGrad can be explained by its spatial contribution aggregation scheme, Frobenius Norm, especially after comparing it with Grad-CAM. In addition, there is only a slight difference among the available settings of NormGrad, except for the bias layer. When the virtual bias layer is placed at the of the network, it is unable to exploit activations while generating the heat-maps. Hence, it generates a constant spatial contribution across the heat-map since the upstream gradient is a single scalar for the last layer. However, this problem can be handled by placing additional virtual bias layers within the intermediate layers of the network. As we combine 4 heat-maps using these virtual bias layers, we are able to increase the accuracy from 0.120 to 0.776 for the validation split, from 0.112 to 0.766 for the testing split of the Object CXR dataset.

### 4. Discussion and Conclusions

In this work, we proposed an automatized framework for medical image quality analysis, for which we used NormGrad to explain the decisions of the framework. We empirically showed that NormGrad provides more accurate saliency maps, which are centered across the target object of interest, after comparing the other saliency methods. Furthermore, the number of false positives falls as a result of using NormGrad instead of Grad-CAM. We also noticed that NormGrad is more robust to model-based misclassification errors after comparing it to the Grad-CAM's saliency maps. We additionally validated these findings quantitatively

Table 2: Pointing Game results on the validation and testing splits of Object CXR comparing the other baseline methods.

| Method | Layer Type | Heat-map | Val | Test |
|---|---|---|---|---|
| NormGrad | Scaling | Single | 0.878 | 0.854 |
| | | Combined | 0.878 | 0.846 |
| | Conv 1x1 | Single | 0.876 | 0.856 |
| | | Combined | **0.880** | 0.846 |
| | Conv 3x3 | Single | 0.874 | **0.862** |
| | | Combined | **0.880** | 0.850 |
| | Bias | Single | 0.120 | 0.112 |
| | | Combined | 0.776 | 0.766 |
| InputxGrad | - | - | 0.246 | 0.240 |
| Guided BP | - | - | 0.208 | 0.188 |
| Guided Grad-CAM | - | - | 0.348 | 0.334 |
| Grad-CAM | - | - | 0.684 | 0.656 |

through the Pointing Game benchmark, where we obtained superior performance after comparing other saliency detection methods on the Object-CXR dataset.

NormGrad achieves significant performance improvements in our qualitative and quantitative assessments. Interestingly, our findings are opposing the claims shown in (Wang et al., 2020) which uses the same accuracy metric presented in this work. However, the performance of NormGrad is also dependent on its configuration, namely, the virtual identity layer choice, if and how the heat-map combination scheme is applied. As a result, NormGrad is a promising method for neural network interpretability, specifically for Chest X-Ray image quality analysis.

As future work, we are interested in assessing the effect of combining the heat-maps of different layers and investigating the re-weighting options for heat-map combination. Besides, we would like to perform further experiments on different image modalities, e.g., CT and MR. Lastly, we would like to enhance the NormGrad framework by providing medical insight.

## Acknowledgments

We thank Mehmet Ozan Unal for his contribution to the development of the pre-trained model. This paper has been produced benefiting from the 2232 International Fellowship for Outstanding Researchers Program of TUBITAK (Project No: 118C353). However, the entire responsibility of the publication/paper belongs to the owner of the paper. The financial support received from TUBITAK does not mean that the content of the publication is approved in a scientific sense by TUBITAK.

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

# Appendix A. Quantitative Results

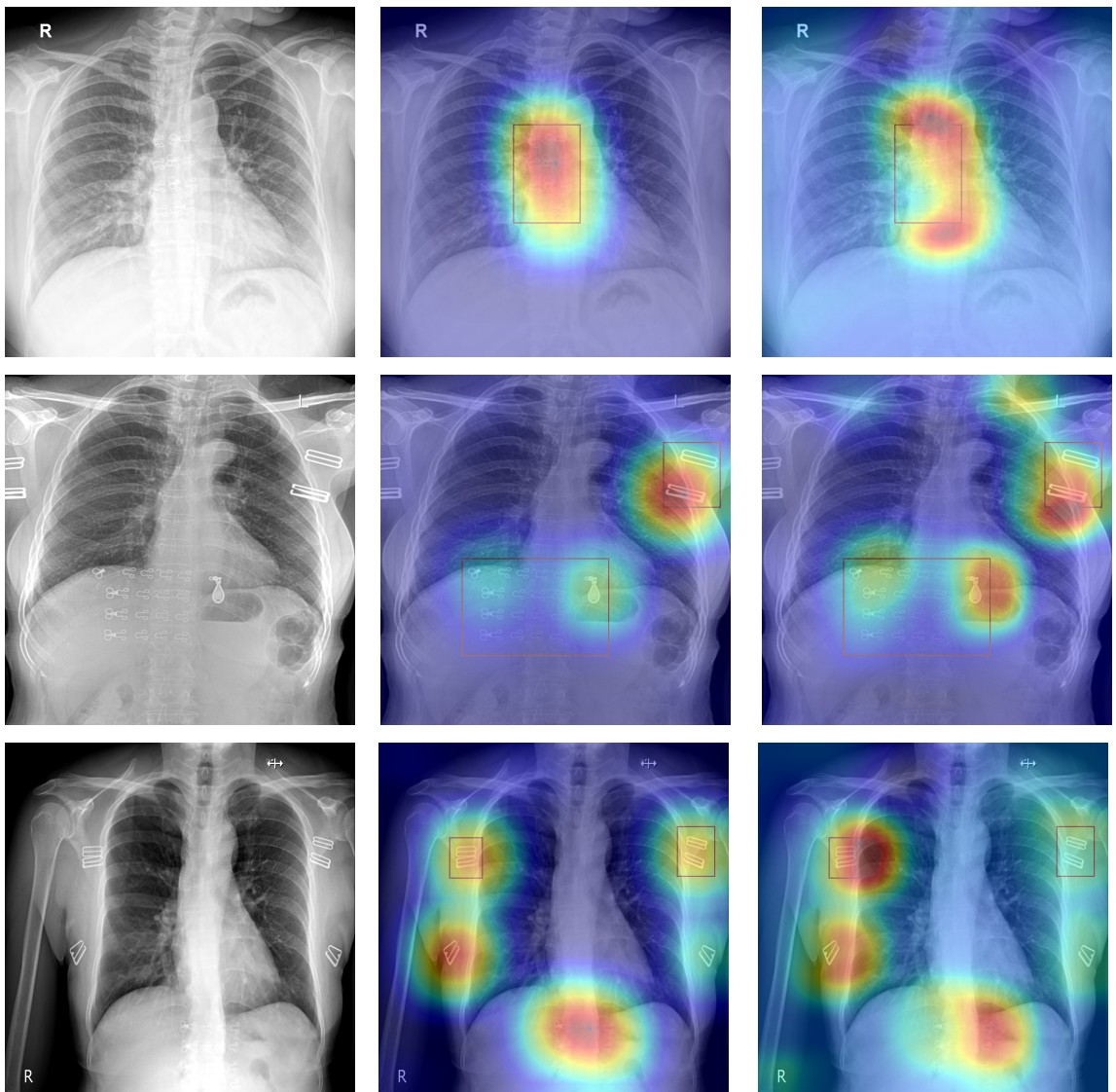

Figure 3: More samples regarding foreign object localization with NormGrad and Grad-CAM. From left to right: Original Image, NormGrad conv3x3 single, Grad-CAM.

## Appendix B. NormGrad Failures

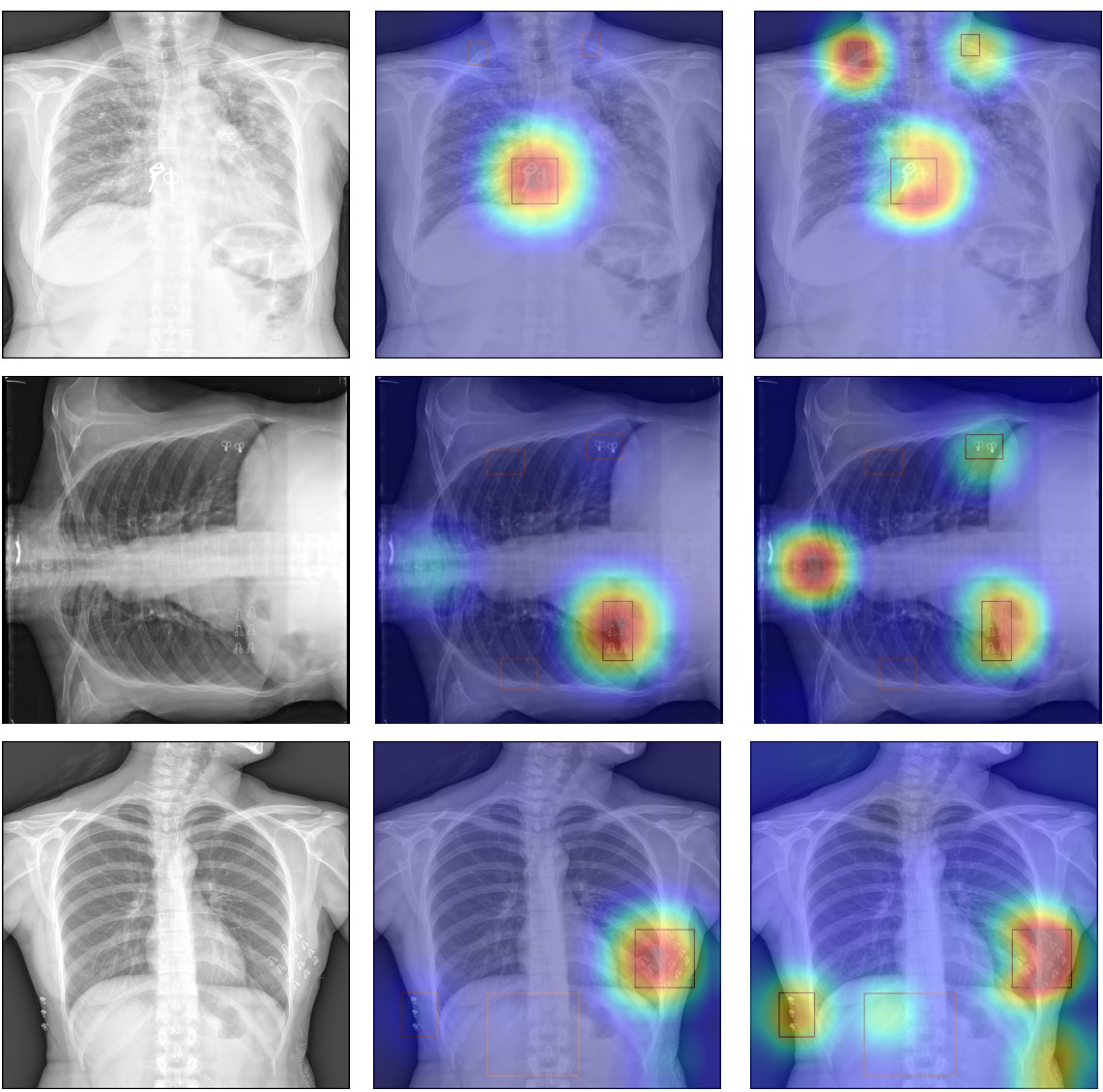

Figure 4: Some sample images where NormGrad fails to detect all the foreign objects. From left to right: Original Image, NormGrad conv3x3 single, Grad-CAM.

## Appendix C. False Positive Comparison

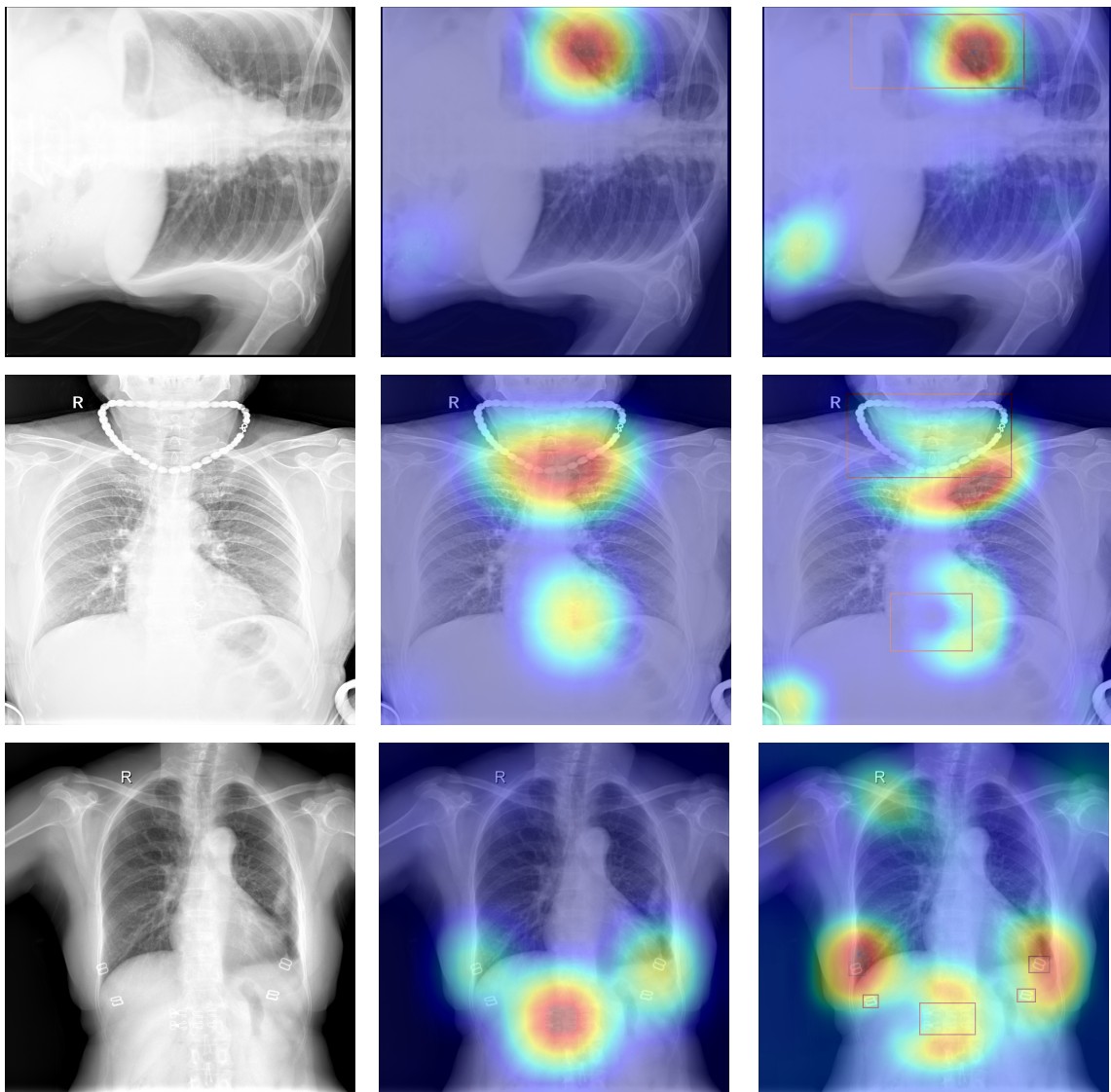

Figure 5: Some sample images where NormGrad has regions with less number of false positives. From left to right: Original Image, NormGrad conv3x3 single, Grad-CAM.

## Appendix D.  Extracging Misclassifications

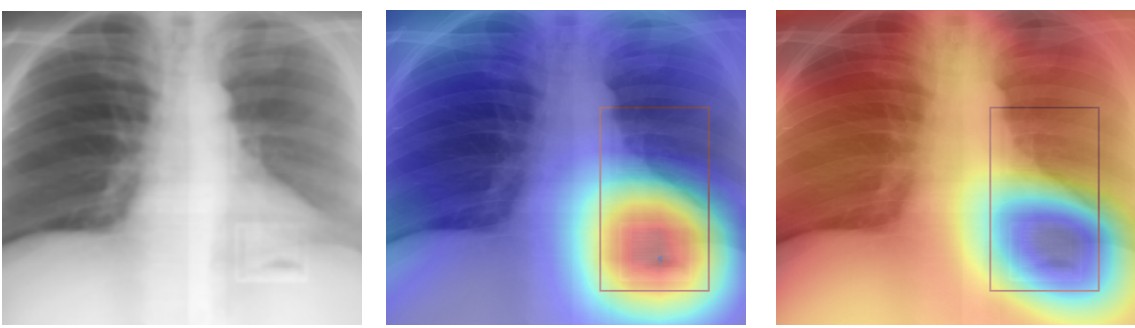

Figure 6:  Attention maps across different saliency and attribution methods when the image is misclassified.  Original Image, NormGrad conv3x3 single, Grad-CAM. Image has been cropped after obtaining saliency maps.

