# OpenReview forum: "Explainable Image Quality Analysis of Chest X-Rays"
_MIDL.io/2021/Conference — MIDL 2021_

### Official Review · AnonReviewer1 · 2021-03-07

**Confidence:** 3
**Preliminary Rating:** 3
**Recommendation:** Poster
**Final Rating:** 3

**Summary:**

This paper explores the application of NormGrad for producing saliency maps in a classifier trained to detect foreign objects in chest x-Rays. The authors show quantitatively that NormGrad significantly outperforms other techniques such as grad-CAM, guided grad-CAM and Input x Grad. The authors also demonstrate in an example that normGrad is less sensitive to misclassifications.

**Strengths:**

1. The results and message of the paper are very clear. The authors intended to demonstrate the superiority of normGrad for IQA in chest x-Rays versus other techniques - this is clearly demonstrated.

2. The paper is generally well written, the literature review is expansive and covers the state of the art well, placing the context of saliency map well and thus motivates the application of norm Grad.

**Weaknesses:**

1. The authors make note of opposing claims to their findings by Wang et al. in Section 4 which is good. However, there is little explanation for why this occurs? What is it about the authors experiments, dataset and configuration which causes this? Further exploration would permit the reader to ascertain the strength and merit of the method for IQA then.

**Deanonymize Review:**

no

**Detailed Comments:**

1. Are there any examples where NormGrad fails significantly or spectacularly? An analysis of the failure modes of the best performing method would be welcome as this might motivate some new incremental changes to the algorithm to improve performance on medical image datasets.

2. As an application paper, I would like to see more visual/qualitative figures of the methods in an Appendix.

3. 1st line page 2 - "Inexplaniability" --> typo

4. I would revise the LaTex math commands. For instance, Section 2, line 6 $x \in \mathbb{R}^{CxWxW}$. I would change the symbol for the $x$ everywhere in the manuscript to something like $x \in \mathbb{R}^{C\times W \times W}$ using "\times" instead of "x"

5. In the discussion, I find the semantics quite loaded. Stating "As a result NormGrad is a promising method,..., specifically for medical image quality analysis". The experiments were on chest x-Ray IQA thus I would specify it is promising for this application. Considering the authors specifically mentioned different results in Wang et al., this claim cannot be made.

6. Why did guided grad-CAM perform so poorly when compared to grad-CAM in Table 2?

**Final Rating Justification:**

I would like to thank the authors for their response and for their updates in the paper. The manuscript is a lot more self-contained now and feels more complete. I am therefore still leaning to accept the paper.

**Justification Of The Preliminary Rating:**

Fairly good validation paper displaying merits of NormGrad for IQA in chest x-Ray scans. Clear experiments, no major faults. For me to give strong accept, I would like to see more extensive evaluation as to where the method fails, more extensive theoretical evaluation compared to Wang et al. who obtained opposite results and recommendations for improving the method specifically on medical imaging datasets.

**Paper Type:**

validation/application paper

**Questions To Address In The Rebuttal:**

1. More explananations on failure modes of NormGrad
2. Evaluation as to why the author's results are opposite to those of Wang et al.


**Special Issue:**

no

---

> ### Author Response · Authors · 2021-03-18
> **Reply to AnonReviewer1**
>
> We thank the reviewer for their comments. We have adopted the above-mentioned suggestions within our text.
>
> **On Weaknesses and Wang et al.:** Although we cannot identify the exact reason why NormGrad did not work in Wang et. al., they mentioned that they only collected data from a single vendor, and hence, they were unable to perform any cross-vendor validation experiments. In addition, their conclusion is mainly based on the fusion of 3DCC (3D Connected Components) and saliency detectors, in which they observed a significant performance in favor of CAM instead of NormGrad. However, we see comparable performance when CAM is compared to NormGrad without using 3DCC as in Table V of their work. Lastly, they applied NormGrad on a different problem setting outside the interests of medical IQA, i.e., COVID-19 Lesion Localization from Chest CT.
>
> **On the Failing Examples:** Although there are few difficult examples where NormGrad demonstrates poor performance compared to Grad-CAM, it is still able to localize one of the prominent objects within the X-Ray image, with less number of false positives. We provided some samples of these circumstances in the appendix section. One interesting challenge as a way for further development of NormGrad would be fusing it with Grad-CAM to effectively leverage both of these methods’ strengths.
>
> **On Semantics in the Discussion section:** We updated the Discussion section according to the task in the experimental results.  As future work, we are planning to proceed with other image domains to generalize its capacity for general-purpose IQA.
>
> **About the Poor Performance of Guided Grad-CAM:** Formally, Guided Grad-CAM is the combination of Grad-CAM and Guided BP in an element-wise multiplication manner. When we observe the results in Table 2, we see that Guided BP has the lowest performance in terms of Pointing Game accuracy for both validation and test splits. Hence, this degradation of performance has been reflected in Guided Grad-CAM.

---

### Official Review · ~Jannis_Hagenah1 · 2021-03-08

**Confidence:** 4
**Preliminary Rating:** 3
**Recommendation:** Poster
**Final Rating:** 4

**Summary:**

In this work, the authors present an approach to integrate explainability in the form of saliency maps into foreign object detection in chest X-rays. As the foreign object detection is used for fast image quality assessment, the authors claim that this study presents the first work on explainable image quality assessment in chest X-ray imaging. Thus, multiple methods for computing saliency maps are applied to a publicly available dataset and several hyperparameters are evaluated. The authors identify that NormGrad outperforms all compared methods with very promising results.

**Strengths:**

The paper has a clear aim and focus, is technically sound and mostly easy to follow. Even though the methods are selected from recent computer vision literature, their application to chest x-ray quality assessment seems to be novel and tackles a relevant clinical problem. The evaluation of hyperparameters as well as the comparison of NormGrad to other methods is sound and extensive, the quantitative results are impressive.

It is worth mentioning that the authors evaluate the methods on an open dataset and that the code developed in the scope of this study will be made publicly available. This is highly appreciated and follows the open science spirit of MIDL!


**Weaknesses:**

One downside of the paper is a lack of qualitative results. Even though the authors extensively discuss their evaluation results qualitatively, only one example image is given. It would be great to see more examples in the appendix, including some worst-cases.

Unfortunately, the mathematical description of GradCam and NormGrad (sections 2.1 and 2.2) are hard to follow. Replacing some text descriptions by equations might clear this up a bit. Please also refer to the “Detailed Comments” section below.

Another issue arises regarding the author’s claim that “this work is the first that uses NormGrad in the medical imaging context” (page 2). In section 4, the authors refer to the recent study of Wang et al. (2020) that also applied NormGrad in the scope of chest X-ray imaging. Hence, the claim should be adjusted and rephrased as it clearly does not hold.


**Deanonymize Review:**

yes

**Detailed Comments:**

Besides the points listed above, I have some minor comments:

-Throughout the whole paper, matrix sizes are given by writing the standard x symbol instead of the $\times$ symbol (using \times). This should be changed in sections 2 and 3 as well as Table 1.

-In 2.1 and especially in 2.2, the quite crowded textual explanation of mathematical operations is a bit confusing and hard to follow. It might be easier to use some equations instead of specific text parts.

-In Fig. 1, it would be great to explain the abbreviations in the caption, e.g. GAP.

-Fig 2. Should be typeset to be at the top or bottom of page. Like this, it is a bit confusing that there is text above the figure. This might be very subjective, but at least in my opinion, it would be nicer somewhere on top.

-In 3.1, you state that you pre-trained the ResNet model on ImageNet and fine-tuned it on the chest X-ray data. Did you replace some layers in the model for the transfer? And how did you manage to transfer from three-channel RGB images to single channel grayscale X-ray? Please clarify this!

-In 3.4, you use the variable names $H$ and $M$ to define your accuracy metric $A$. However, $H$ and $\mathbf{M}$ were already in use in section 2. It would be good to rename the variables here to achieve a consistent notation.

-In 4, you state that “the number of false positives falls as a result of using NormGrad instead of Grad-CAM”. Without an appropriate number of qualitative examples given somewhere in the paper, e.g. the appendix, this claim lacks of a sound basis. Please provide some examples to proof your claim. This is related to my general comment on more qualitative results (see above).


**Final Rating Justification:**

The authors provided a revised manuscript of a significantly increased quality. Specifically, the description of NormGrad is more clear and a broader set of qualitative results is provided. The authors addressed all my concerns and comments in an adequate way. Hence, I am happy to vote for a strong accept of this nice work.

**Justification Of The Preliminary Rating:**

The presented study tackles a relevant problem with a state-of-the-art methodology, a sound evaluation, promising results and an all-in-all clear presentation. The approach is original and the results are of high interest for clinicians as well as for researchers in the area of image quality assessment. Additionally, the code will be publicly available and the results will be reproducible due to the usage of open data. Hence, I am happy to vote for accepting the manuscript while highly encouraging the authors to submit a revised version with some clarifications (see above). I am sure that the revised version will be a nice paper of great interest. Congratulations on this work!

**Paper Type:**

validation/application paper

**Questions To Address In The Rebuttal:**

As mentioned above, my main concerns do not relate to the methods or the results itself but on the presentation. Please add more qualitative results, revise section 2 and reformulate your contribution (see above). If you also address all my detailed comments, I am sure that the revised manuscript will be a great paper and am willing to increase my vote to a strong accept.

**Special Issue:**

no

---

> ### Author Response · Authors · 2021-03-18
> **Reply to AnonReviewer3**
>
> We thank the reviewer for her/his motivating and insightful comments.
>
> **On Weaknesses:** We included several samples, which some of them are the worst-case scenario for NormGrad, in the appendix. We also revised section 2 for improving the quality of the manuscript. Lastly, we updated the manuscript highlighting that we are the first paper that utilized NormGrad on medical IQA.
>
> **Lack of Qualitative Results:** We included several samples, which some of them are the worst-case scenario for NormGrad, in the appendix.
>
> **Insufficient clarity of Methods Section:** We revised section 2 for improving the quality of the method description.
>
> **First paper on Normgrad:** We updated the manuscript highlighting that we were the first paper that utilized NormGrad on medical IQA. Moreover, our results indicate the superior performance of NormGrad in comparison with Grad-CAM and other baselines.
>
> **Minor comments:** We thank the reviewer for the comments on the $\times$ symbol, math, Figure captions and positioning, ResNet pre-training. We have updated the manuscript according to each comment. H and M notations are changed as H → T, M → F.  We have added some examples in the Appendix, which illustrates the comment about the reduced number of false positives.

---

### Official Review · AnonReviewer4 · 2021-03-08

**Confidence:** 4
**Preliminary Rating:** 2
**Final Rating:** 3

**Summary:**

The authors evaluated an approach for explainable image analysis of foreign objects classification in chest X-ray images. The explainability of the approach relies on NormGrad, a method that can localize foreign objects with saliency maps of the classifier. The approach was compared with a few other saliency detection methods and has been evaluated qualitatively and quantitatively using a Pointing Game accuracy, showing advantage to the other baselines.

**Strengths:**

- The paper has successfully applied a recent saliency detection method NormGrad in the context of medical image analysis which shows promising results.
- The evaluation was carried out on a fairly large dataset and includes comparison with a few other baseline methods both qualitatively and quantitatively which show advantages.

**Weaknesses:**

- Insufficient clarity in the description and the discussion of the method. I need to read the original paper in the reference (Rebuffi et al. 2020) to get a more comprehensive understanding of the approach. Please try to address the following points more clearly:
	- What's the motivation of using a virtual identity layer? Why does it help?
	- What's the motivation and benefit of using the Frobenius norm? How does the Frobenius norm explain the merit of NormGrad?
	- What's the difference between the unfold and the flatten operation (Figure 1)?
	- When combining multiple heatmaps, were the same type of virtual identity layers used to compute them? What's the motivation of using the geometric mean to combine them?
- Insufficient clarity in the evaluation of the method
	- Dataset: in the validation and testing set, how many samples are with foreign objects, how many samples not?
	- Pointing Game accuracy:
		- To compute this metric, was the location of the maximum value in the whole saliency map checked, or the locations of the maximum value of each attended area in the saliency map? How does it deal with the situation where there are multiple foreign objects in the image?
		- what value was set to the offset to the bounding box annotation in the experiment?
	- Table 2: please explain the difference of accuracy on the setting Bias Single and Bias Combined.
- Insufficient validation of the method
	- The problem setting in this paper is binary classification (with or without foreign objects in an image). Since an image can contain multiple foreign objects, it would also be interesting to know the precision and recall of the computed saliency maps. Such as how many true foreign objects are missed? How many false positive detections in an image? For example in Figure 2, some foreign objects are not detected, which makes me question the explainability of the method. Why does it attend some foreign objects, but not some others?

**Deanonymize Review:**

no

**Detailed Comments:**

- When denoting the shape of an array in the math notation, please use the multiplication sign between the dimensions instead of the letter "x".
- In Figure 3, since the classifier has not predicted any foreign object in the image (which is a positive sample), and NormGrad still shows a positive spot in the image, how does it explain the model failure in this example?
- I find the term "Image quality analysis" in the title and paper too generic. Bad image quality can mean various things, such as noise, artifacts, etc. The papers focus only on detection of foreign objects which doesn't touch other image quality issues. It would be better to make the topic more specific and clear to the readers.

**Final Rating Justification:**

I appreciate the authors for answering my questions and updating the paper. Most of my questions have been addressed and the updated manuscript makes the description of the method clearer and self-contained, which improves its readability. Although, as a validation paper, I think it would still be good to evaluate more thoroughly on the method, e.g. giving quantitative evaluation on the false positive / negative detection of the method to provide more insights to the explainability, weighing on the strengths and the weaknesses, I'm lean to accept the paper.

**Justification Of The Preliminary Rating:**

Although the authors have introduced the NormGrad method for explanation of foreign object classification and show some promising results, the insufficiency in the clarity and validation of the paper weakens the strengths.

**Paper Type:**

validation/application paper

**Questions To Address In The Rebuttal:**

Please address the questions in the weaknesses and detailed comments.

**Special Issue:**

no

---

> ### Author Response · Authors · 2021-03-18
> **Reply to AnonReviewer4**
>
> We thank the reviewer for her/his insightful feedback and detailed comments. We have revised Section 2.2 to address your questions regarding the motivation of the methodology while ensuring the text being self-contained.
>
> **About description and the discussion of the method:** We described our motivation of using virtual identity layers and Frobenius norm, and defined the unfold operation in-text. To reiterate our motivation to use virtual identity and Frobenius norm, we would like to exploit the activations and gradients in as many ways as possible and to be invariant to the shape of spatial contributions, specifically as of a convolutional layer. On the last inquiry regarding the use of geometric mean to combine the saliency maps, we could have also used arithmetic mean or any other way that was mentioned in Rebuffi et. al. to calculate a unified heat-map; however, we are yet to perform further analysis on the heat-map combination settings.
>
> **The clarity in the Evaluation of the Method:**  Pointing Game uses the global maximum value of the saliency map while computing the accuracy. When there exist multiple objects, checking the closeness to either one of the foreign objects is sufficient to determine whether it is a hit or a miss. We would like to evaluate how to improve this metric even further. Although we defined $\tau=25$ as our default value within the scope of these experiments, we also set $\tau=15$ to observe the changes in performance specifically for Grad-CAM and single and combined settings of NormGrad (for the convolutional layers only) comparing when $\tau$ is set to 25.
>
> | Method | Val | Test |
> | --------------| ----| ------ |
> | NormGrad, conv1x1, single | 0.872 (-0.004) | 0.848 (-0.008) |
> | NormGrad, conv1x1, combined| 0.878 (-0.002) | 0.840 (-0.006) |
> | NormGrad, conv3x3, single | 0.866 (-0.008) | 0.850 (-0.012) |
> | NormGrad, conv3x3, combined| 0.876 (-0.004) | 0.846 (-0.004) |
> | Grad-CAM | 0.640 (-0.044) | 0.630 (-0.026) |
>
> **Multiplication signs:** We thank the reviewer for the comment, in the updated manuscript we have updated the multiplication signs.
>
> **Fig.3 Wrong Classification and Correct Explanation:** We thank the reviewer for pointing out a very interesting observation of the explainability capability of the method, even when the final binary classification is not accurate. The method is capable of illustrating the correct localization of the network for the classification task even though the final classification output is wrong. The final classification is done using 0.5 as a threshold for the binary classification and the example in Fig.3 is a particular example, where the probability of detecting a foreign object was close to the threshold value. We have added additional examples of the Appendix to illustrate the advantages and shortcomings of Normgrad.
>
> **Inappropriate Title:** We thank the reviewer for pointing out the general nature of the title. At this point, we are unable to update the title, but we have included detailed information in the abstract on the specific task (foreign object detection) we focused on to avoid confusion.

---

### Official Review · AnonReviewer2 · 2021-03-08

**Confidence:** 4
**Preliminary Rating:** 4
**Recommendation:** Oral
**Final Rating:** 4

**Summary:**

The authors present an explainable deep learning system for image quality assessment based on the detection of foreign objects in chest x-ray data together with an explainable pipeline using NormGrad. The paper qualitatively and quantitatively validates the improved localisation properties of their approach.

**Strengths:**

- The authors address a very relevant problem for medical applications of deep learning. To my knowledge this is a quite unique approach.
- The authors show convincing qualitative and quantitative evidence.

**Weaknesses:**

- I felt that the explanations in 2.2. NormGrad were not really sufficient to understand the proposed approach (in particular without the original paper describing NormGrad itself). I would strongly recommend revising this part and make it easier to follow and more self-contained.
- Maybe the authors could add some thoughts about how easily this would generalise to related examples or use cases.


**Deanonymize Review:**

no

**Final Rating Justification:**

My major point has been addressed by the authors. I think the explanation of the method in section 2 is much better in the revised version of the manuscript.

**Justification Of The Preliminary Rating:**

I think the topic is very relevant and to the best of my knowledge, this paper presents a new approach for explainable image quality assessment of x-ray images. The paper seems technically sound and the results are convincing.

**Paper Type:**

validation/application paper

**Questions To Address In The Rebuttal:**

- revise Methods part

**Special Issue:**

yes

---

> ### Author Response · Authors · 2021-03-18
> **Reply to AnonReviewer2**
>
> We sincerely appreciate your review and helpful comments. We have updated Section 2 for clarity. In addition, we would like to extend this methodology to various image quality problems, e.g., cardiac and brain MRI IQA issues.

---

### Author Response · Authors · 2021-03-18
**Comment to all reviewers**

We thank all the reviewers for their constructive feedback to make this manuscript as great as possible. We revised the manuscript that critically important changes are marked in red font in the revised manuscript. To highlight these changes:
* We edited Section 2 to be self-contained and to include our motivation of using components, such as the virtual identity layer and Frobenius Norm.
* We modified our claim to be the first to use NormGrad in the medical image quality analysis.
* We explained the reason why placing a bias layer at the end of the network ends up in poor performance in the Pointing Game benchmark.

In addition, we performed a small experiment to measure the effect of decrementing $\tau$ from 25 to 15, which is to observe the changes in the proximity of the peak point of saliency maps to the bounding boxes.

| Method | Val | Test |
| --------------| ----| ------ |
| NormGrad, conv1x1, single | 0.872 (-0.004) | 0.848 (-0.008) |
| NormGrad, conv1x1, combined| 0.878 (-0.002) | 0.840 (-0.006) |
| NormGrad, conv3x3, single | 0.866 (-0.008) | 0.850 (-0.012) |
| NormGrad, conv3x3, combined| 0.876 (-0.004) | 0.846 (-0.004) |
| Grad-CAM | 0.640 (-0.044) | 0.630 (-0.026) |

---

### Meta-Review · Area_Chairs · 2021-03-29

**Recommendation:** Accept (Oral)

**Metareview:**

The paper receives unanimously positive reviews from four knowledgeable experts. They all agree that, though the saliency detection method NormGrad is not invented by the authors, its use in the context of medical image analysis shows promising results, which is clearly demonstrated by the authors. They also express some concerns, which are largely addressed by the authors in the discussions. I, therefore, recommend the acceptance of this paper.

**Paper Type:**

validation/application paper

---

### Decision · Program_Chairs · 2021-03-31

**Decision:**

Accept

**Comment:**

Congratulations your paper has been selected as a long oral.